# Repurposing the Antiplatelet Agent Ticlopidine to Counteract the Acute Phase of ER Stress Condition: An Opportunity for Fighting Coronavirus Infections and Cancer

**DOI:** 10.3390/molecules27144327

**Published:** 2022-07-06

**Authors:** Anna Tesei, Michela Cortesi, Martina Bedeschi, Noemi Marino, Giacomo Rossino, Roberta Listro, Daniela Rossi, Pasquale Linciano, Simona Collina

**Affiliations:** 1Biosciences Laboratory, IRCCS Istituto Romagnolo per lo Studio dei Tumori (IRST) “Dino Amadori”, 47014 Meldola, Italy; michela.cortesi@irst.emr.it (M.C.); martina.bedeschi@irst.emr.it (M.B.); noemi.marino@irst.emr.it (N.M.); 2Medicinal Chemistry and Pharmaceutical Technology Section, Department of Drug Sciences, University of Pavia, 27100 Pavia, Italy; giacomo.rossino@unipv.it (G.R.); roberta.listro01@universitadipavia.it (R.L.); daniela.rossi@unipv.it (D.R.); pasquale.linciano@unipv.it (P.L.)

**Keywords:** sigma 1 receptor (S1R), endoplasmic reticulum (ER), ER stress, terminal UPR, P2Y_12_ inhibitors, ticlopidine, COVID-19, cancer

## Abstract

Different pathological conditions, including viral infections and cancer, can have a massive impact on the endoplasmic reticulum (ER), causing severe damage to the cell and exacerbating the disease. In particular, coronavirus infections, including SARS coronavirus-2 (SARS-CoV-2), responsible for COVID-19, cause ER stress as a consequence of the enormous amounts of viral glycoproteins synthesized, the perturbation of ER homeostasis and the modification of ER membranes. Therefore, ER has a central role in the viral life cycle, thus representing one of the Achilles’ heels on which to focus therapeutic intervention. On the other hand, prolonged ER stress has been demonstrated to promote many pro-tumoral attributes in cancer cells, having a key role in tumor growth, metastasis and response to therapies. In this report, adopting a repurposing approach of approved drugs, we identified the antiplatelet agent ticlopidine as an interferent of the unfolded protein response (UPR) via sigma receptors (SRs) modulation. The promising results obtained suggest the potential use of ticlopidine to counteract ER stress induced by viral infections, such as COVID-19, and cancer.

## 1. Introduction

Accumulating evidence shows that cells infected with coronaviruses exhibit increased expression of several proteins correlated with endoplasmic reticulum (ER) stress [1,2]. In particular, approximately 40% of SARS-CoV-2 interacting proteins were recently associated with endomembrane compartments, including ER [3]. Coronavirus infection has a massive impact on the ER due to the enormous amounts of viral glycoproteins synthesized, leading to the perturbation of ER homeostasis. Moreover, virus infection modifies and exploits ER membranes: double-membrane vesicles (DMVs), the coronavirus RNA synthesis site and virus envelopes are derived from the ER membrane [4,5], and, at the end of the replication and assembly cycle, virions bud from the ER-Golgi intermediate compartment [5,6]. These observations highlight the central role of the ER in the viral life cycle and suggest that the ER may represent one of the Achilles’ heels on which therapeutic intervention can be focused.

Despite the availability, with unprecedented rapidity of vaccines that proved effective in preventing and protecting against infection from SARS-CoV-2, the management of the complications that can arise in the course of this viral illness remains challenging. In fact, COVID-19 causes a pulmonary coagulopathy that significantly contributes to worsening the clinical picture of the disease, with consequent severe impact on cardiovascular and cerebral adverse events [7,8]. Of note, starting from its outbreak in late 2019, COVID-19 has caused over 500 million confirmed cases and over six million deaths globally [9]. In this context, immuno-compromised subjects and patients with other co-morbidities (e.g., cancer) represent a particularly vulnerable population that either cannot undergo vaccination or that shows lower immune response against SARS-CoV-2 upon vaccination. Finally, the frequency of the emergence of new virus variants is of major concern for the potential resistance toward the currently available vaccines. In particular, the frequency with which highly virulent coronavirus strains have emerged highlights, as an additional need, the identification of targets for broad coronavirus inhibitors. Since families of viruses often exploit common cellular pathways and processes, targeting the host proteins essential for viral replication may represent a successful strategy that can avoid resistance and lead to therapeutics with broad-spectrum activity. For these reasons, the identification of additional therapeutics and the widening of treatment options remain of great importance [10,11].

The recent works by Gordon et al. [3] suggest the viability of targeting the host–virus interaction network to develop new and effective therapies against coronavirus infections. Particularly, in the case of SARS-CoV-2, the authors identified 66 druggable human protein or host factors targeted by currently approved drugs. The repurposing of approved drugs to target the host–virus interface can represent a viable and time-effective strategy to develop additional therapeutic options, potentially useful when vaccines and virus-targeted drugs cannot be used and/or result ineffective. Notably, in the work by Gordon and collaborators, the drug screening by multiple viral assays initially showed anti-COVID-19 effectiveness just for the mRNA translation inhibitors and for sigma-1 and sigma-2 receptors modulators (S1R and S2R, SRs) [3]. Gordon et al., in a later article, identified shared biology and potential drug targets among the three highly pathogenic human coronavirus strains, i.e., SARS-CoV-2, SARS-CoV-1 and MERS-CoV, identifying 73 host factors that, when depleted, caused significant changes in SARS-CoV-2 replication. From this list of potential drug targets, they validated the biological and clinical relevance of SR1, making it a potential candidate for broad-spectrum treatment of coronavirus infections [12]. S1R exerts its main activity in ER as a gatekeeper of ER stress, which is precisely due to misfolded proteins accumulating above a critical threshold. ER stress induces a series of signal transduction cascades, named unfolded protein response (UPR), aimed to reestablish ER proteostasis by recovering the folding capacity of the cells. However, suppose the adaptive responses fail to restore protein-folding homeostasis. In that case, UPR signaling continues to persist and eventually morphs into alternate signaling programs, called ‘‘terminal UPR’’, that ultimately promote apoptosis [13,14]. ER stress and sustained UPR signaling are significant contributors to the pathogenesis of several diseases, including inflammatory disorders, viral infections, neuronal degeneration and brain injury, and can increase these events’ severity [15,16].

The idea that the triggering of terminal UPR may be harmful to the progress of virus infection originated from the evidence of the extensive use of the ER membrane by a coronavirus. Modulation of UPR may be useful in counteracting the pulmonary inflammatory response and microvascular diseases in severe COVID-19. Based on our longstanding experience regarding SRs modulators and prompted by the medical need related to the COVID-19 pandemic, we directed our efforts toward a drug repurposing approach, which offers reduced risk of failure and a time frame for drug identification to treat COVID-19 patients. Unlike virus-targeted antiviral agents (such as remdesivir and lopinavir, already tested in numerous clinical trials), we focused our attention on drugs able to address adaptive cellular mechanisms of host cells crucial for viral infection by targeting S1R [17,18]. 

Regarding their potential antiviral activity, some drugs endowed with S1R affinity (i.e., amiodarone, amodiaquine, benztropine, chloroquine, chlorpromazine, clemastine, clomipramine, cloperastine, fluphenazine, haloperidol, HCQ, loperamide, mefloquine, methdilazine, progesterone, promethazine, siramesine, tamoxifen, terconazole, thiethylperazine, toremifene and verapamil) have been identified in in vitro drug repurposing screens aiming to identify antiviral compounds against SARS-CoV-2 [3,19,20,21]. Herein, we extended the repositioning approach, investigating the capability of other selected FDA-approved drugs able to bind SRs to counteract the ER stress induced by SARS-CoV-2 infection.

## 2. Results and Discussion

The quinoline-derived drug HCQ (reported S1R *K_i_* = 125.89 nM) [3] has mainly been used for the treatment of malaria but also for rheumatoid arthritis, lupus, porphyria cutanea tarda [22] and was promptly proposed for treating COVID-19 patients [23,24]. However, in June 2020, the FDA revoked the emergency use authorization to use hydroxychloroquine to treat COVID-19 since it does not provide benefits in terms of decreasing death or speeding recovery (FDA communication, 2020). The mechanism of action proposed for HCQ is related to its ability to increase the endosomal pH and to inhibit lysosomal enzymes, thus preventing viruses from entering the host cells [25]. Only recently, the contribution of S1R to the antiviral effects of HCQ has been proposed [3,19]. Conversely, ticlopidine is an approved antiplatelet aggregation drug never presented so far to treat SARS-CoV-2 viral infection. It is a prodrug that is metabolized to the active form, which blocks the ADP-receptor-involved platelet aggregation [26]. Besides, ticlopidine binds S1R as such, with a *K_i_* of 175 nM [27,28]. 

In this brief report, we describe our results on ticlopidine’s capability in triggering the UPR response to shed light on its capacity to simultaneously contrast SARS-CoV-2 infection and, according to its primary therapeutical indication, to reduce the associated coagulopathy and microthrombus formation. For comparison purposes, we used the sigma ligand hydroxychloroquine, which has already been used in experiments in treating COVID-19 patients. As the first step of this study, we perform molecular docking studies to confirm the interaction at a molecular level of HCQ and ticlopidine itself with the S1R binding site (Figure 1).

Keeping in mind that antiplatelet agents, such as ticlopidine, may decrease tumor growth and metastatic potential, as well as improve survival of cancer patients, we used cancer cells as a cellular model, specifically A549 non-small cell lung cancer (NSCLC), LNCaP prostate cancer, U87 glioblastoma and Panc-1 pancreatic adenocarcinoma cancer cells. These cell lines are representative of both the most common and orphan tumors.

We found that both drugs can induce the expression of UPR effector proteins (Figure 2). Ticlopidine, probably due to its higher affinity for the S1R, induced in all tested lines the expression of binding immunoglobulin protein (BIP), one of the leading partners of the S1R through which S1R actively regulates the calcium ion fluxes at the level of the reticulum-mitochondria associated membranes (MAM) [29]. Conversely, hydroxychloroquine did not significantly affect the expression of chaperone protein BIP. Regarding CHOP and ATF4, both necessary to induce cell death and theoretically interrupt the virus reproduction cycle, only ticlopidine always induced both transcription factors’ expression independently from the tested cell line. Accordingly, the cytotoxic assays showed that ticlopidine exerts cell-killing properties starting from 48 h of exposure regardless of the cell line tested (Figure 3). 

Ticlopidine exerts greater cytotoxicity in the LNCaP cell line in respect to the U87 cell line, representative of glioblastoma multiforme, a type of aggressive brain tumor largely resistant to current treatments. Ticlopidine appears to circumvent inherent apoptosis resistance of U87 cells by inducing terminal UPR, even if to a lesser extent with respect to LNCaP cells. This result is in line with the high sensitivity of LNCaP cells to endoplasmic reticulum (ER) stressors, which is reported in the literature [30].

The promising results obtained suggest ticlopidine’s potential in the treatment of COVID-19 patients, with a particular focus on oncological patients, which belong to the aforementioned vulnerable population. We cannot forget that SRs modulators are under investigation for cancer treatment, even if the mechanisms underlying their antitumor properties are intricate and not yet fully understood. To cope with the acute phase of the ER stress condition, either induced by cancer or viral infections, the cells exploit UPR pathways, and SRs seem to play a pivotal role in UPR modulation. The early and protracted permanence of UPR and subsequent onset of “terminal” UPR sustained by SRs modulators could very quickly lead to the death of the cells before they have the time to adopt a more malignant phenotype or allow the viral life cycle completion. Thienopyridine derivatives with antiplatelet activity, including ticlopidine, are widely prescribed for patients with cardiovascular diseases [31,32]. In addition to their antiplatelet activity being closely related to the inhibition of P2Y_12_ receptor, the literature data highlighted their potential anticancer and antimetastatic properties [33,34,35,36]. Hemostasis and inflammation are intimately linked, inducing and amplifying each other, and this interconnection contributes to many pathological situations, including sepsis, acute lung injury, autoimmune diseases, tumorigenesis and metastasis.

Based on the data reported in the present work, we hypothesized that S1R inactivation by ticlopidine induces a very fast accumulation of unfolded proteins in ER lumen, triggering the terminal UPR response. Actually, the protein chaperone BiP and the transcription factors ATF4 and CHOP belong to the same pathway. Under normal physiological conditions, BiP is associated with ER sensor PERK, present in the ER membrane in an inactive state. As a response to the accumulation of unfolded or misfolded proteins within ER lumen, BiP dissociates from PERK, which becomes an active kinase with the ability to phosphorylate α subunits of the eIF2. The result is suppression of global protein translation, cell cycle arrest in the G1 phase as well as the induction of preferential translation of ATF4, which upregulates the expression of genes responsible for restoring cell homeostasis. Under prolonged ER stress, ATF4 triggers pro-apoptotic signals through induction of CHOP, which is responsible for initiation of the apoptotic cascade [37]. This evidence highlights once again the crucial role of the S1R in controlling ER stress and the activity of ticlopidine as an ER stressor [14,38].

The capability of ticlopidine to bind as such to S1R, to directly modulate the receptors and UPR pathways, had never been described previously and could be crucial in counteracting both cancer progression and COVID-19 infection. Obviously, additional work is needed for clarifying the role of S1R in SARS-CoV-2 infection. Nevertheless, between the completion of the ongoing vaccination campaign and the discovery of ad hoc treatment for COVID-19 patients, the repurposing of the FDA-approved drug ticlopidine may represent a good opportunity in the fight against SARS-CoV-2 infection.

## 3. Materials and Methods

### 3.1. Molecular Docking

The X-ray crystallographic structure of human S1R in complex with the agonist (+)-pentazocine (PDB ID: 6DK1) [39] was used as the starting point for molecular docking calculations. The protein was prepared using the Schrödinger suite Protein Preparation Wizard, and the main structural issues were addressed (e.g., assignment of bonds, bond orders and the addition of hydrogen atoms) [40]. The 3D structures of the compounds were generated from SMILE strings and prepared with LigPrep utility from the Schrödinger suite [41]. Ionization states and tautomers were generated at pH 7.4 ± 0.5 using Epik [42]. Docking calculations were performed with Glide software using the Standard Precision (SP) protocol default settings [43]. The docking protocol was validated by redocking the (+)-pentazocine into the parent receptor, providing a root-mean-square deviation (RMSD) value lower than 2.0 Å. Finally, the prepared ligands were docked into the model, and the resulting ligand–protein complexes were visually inspected.

### 3.2. RealTime RT-qPCR

TRIzol reagent (Life Technologies, Carlsbad, CA, USA) was used, following the manufacturer’s instructions, for total cellular RNA extraction. RNA was quantified by the Nanodrop MD-1000 spectrophotometer system. iScript cDNA Synthesis kit (Bio-Rad Laboratories, Hercules, CA, USA) was used to perform reverse transcription reactions. Further, 400 ng of total RNA was retrotranscribed in 20 µL of nuclease-free water. Real-time PCR was performed by a 7500 Fast Real-Time PCR system (Applied Biosystems, Foster City, CA, USA), and the expression of BIP, ATF4 and CHOP genes were detected by TaqMan assays. Reactions containing 40 ng of cDNA template, TaqMan Universal PCR Master Mix (2X) and selected TaqMan assays (20X) were carried out in triplicate at a final volume of 20 L. Samples were maintained at 50 °C for 2 min, then at 95 °C for 10 min followed by 40 amplification cycles at 95 °C for 15 s and 60 °C for 30 s. The changes in gene expression after drug exposure were displayed relative to the untreated control sample. Gene expression was normalized with two endogenous reference genes, GAPDH and HPRT, identified as the most stable genes by the geNorm VBA applet for Microsoft Excel. Data are reported as mean ± SD. Data were processed using the GraphPad Prism program (version 9.2) (GraphPad Software).

### 3.3. SRB Assay

Cytotoxicity assays were performed using Sulforhodamine B (SRB assay) according to the method by Skehan et al. (1990) [44]. Cells seeded onto a 96-well plate were exposed to increasing concentrations of the drugs. The treatment’s effect was evaluated after 24 h, 48 h and 72 h exposure. Two independent experiments were performed in octuplicate. The optical density (OD) of treated and untreated cells was determined at a wavelength of 540 nm using a fluorescence plate reader. Data were processed using the GraphPad Prism program (version 9.2) (GraphPad Software, San Diego, CA, USA).

## 4. Conclusions

Accumulating evidence suggests the primal role of ER stress and UPR response in several pathological conditions, including viral infections and cancer. Considering that S1R has been identified as a key regulator of both ER stress and UPR, and also as part of the SARS-CoV-2 interactome, in the present work, we investigated two compounds, i.e., HCQ and ticlopidine, selected among FDA-approved drugs able to bind SRs. Their ability to stimulate a UPR response has been evaluated in different cancer cell lines, and ticlopidine resulted in being more effective in inducing the expression of UPR effector proteins (i.e., BIP, ATF4 and CHOP). 

Taking into account the involvement of SRs in cancer, we expanded our study with MTT assays. These showed that ticlopidine exerts cytotoxic effects after 48 h of exposure in all the cancer cell lines investigated. 

These results support the idea that ticlopidine can counteract ER stress by modulating a UPR response via S1R interaction, with potential application in both antiviral therapies and cancer. Hence, ticlopidine may be effective in counteracting COVID-19 infection by acting at the host-virus interface, representing a new and valuable candidate for repurposing. Notably, the exploitation of an antiplatelet drug might represent a better opportunity compared to other approved drugs (i.e., antipsychotics) previously identified for the same purpose. The investigation of our in-house library of S1R modulators [45,46,47] is ongoing, and the results will be reported in due course.

## Figures and Tables

**Figure 1 molecules-27-04327-f001:**
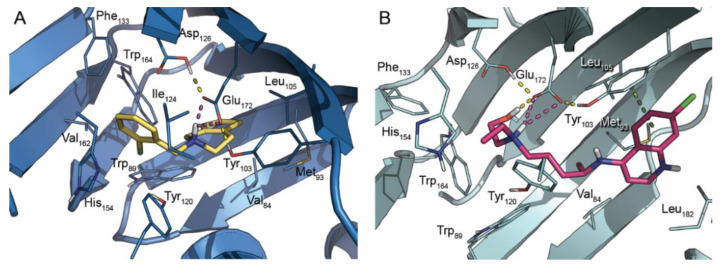
Docking of ticlopidine ((**A**), stick, yellow carbon) and HCQ ((**B**), stick, purple carbon) into S1R binding site. The reference crystal structure of S1R used for the docking calculation (PDB ID: 6DK1) is shown in blue (**A**) and aquamarine (**B**) cartoon. Important interacting residues are in stick representation. Model atoms, except for carbons, are color-coded: oxygen (red), nitrogen (blue) and sulfur (yellow). H-bonds, bridge salt and π−π interactions are represented as yellow, magenta and green dotted lines, respectively. Part of the β-barrel has been hidden for more precise visualization of the binding site.

**Figure 2 molecules-27-04327-f002:**
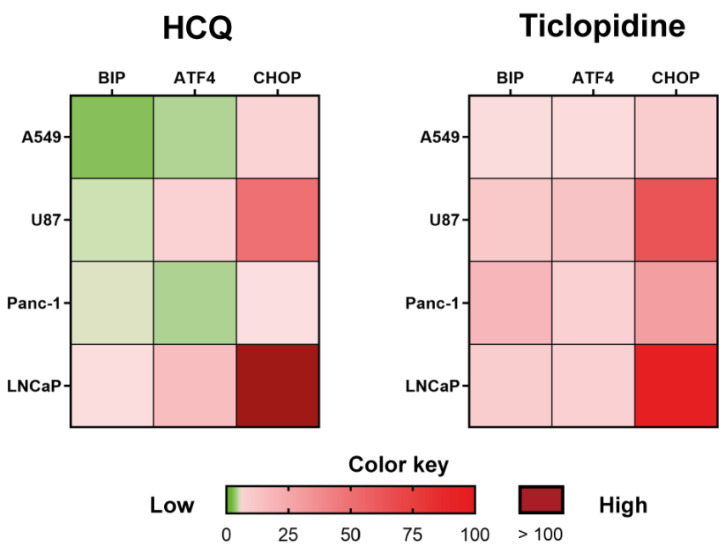
Analysis of UPR-related gene expression. Heatmaps displaying change in mRNA expression levels of three selected UPR markers in 4 different tumor cell lines 24-h exposed to HCQ or ticlopidine 500 µM. Each row represents a cell line, and each column a gene. Red and green colors indicate the level of up- and down-regulation, respectively. Data were processed using the GraphPad Prism program (version 9.2) (GraphPad Software, San Diego, CA, USA).

**Figure 3 molecules-27-04327-f003:**
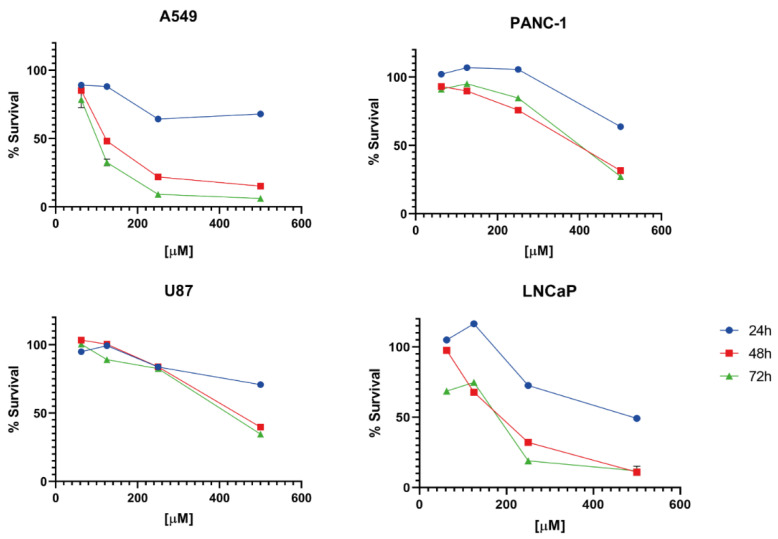
Cytotoxicity exerted by ticlopidine in different cancer cell lines. The cell survival was evaluated at the ticlopidine concentrations of 62.5, 125, 250 and 500 µM, and at 24, 48 and 72 h after drug exposure. Data are reported as mean ± SD.

## Data Availability

The data presented in this study are available within the article.

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
