# Peer review of "Repurposing the Antiplatelet Agent Ticlopidine to Counteract the Acute Phase of ER Stress Condition: An Opportunity for Fighting Coronavirus Infections and Cancer"

_molecules, 2022, doi:10.3390/molecules27144327_

Round 1

Reviewer 1 Report

Under the impact of COVID-19, the number of confirmed cases and deaths have seriously affected the life quality of people all around the world. In addition, the uncertainty of the complications of COVID-19 has stimulated the urgent needs for alternative therapeutics and options against Coronavirus infections. This manuscript aimed to identify the possible application of ticlopidine to counteract viral infection as well as cancer. It is a valuable manuscript since it is well-organized and well-written. Therefore, I recommend the editor to accept this manuscript for publication after addressing the following comments.

1. The authors used hydroxychloroquine (HCQ) as a positive control; however, it has been revoked to treat COVID-19 by the U.S. FDA. Why the authors did not consider different candidates such as chloroquine, which also has endowed with S1R affinity.

2. In Fig 2, the result displayed that ticlopidine obviously induced ATF4 and CHOP in both U87 and LNCaP cancer cells. However, the survival rate of U87 cells was the highest among the four tested cancer cells after the treatment of ticlopidine. Both data seems to contradict each other.

Author Response

  1. The authors used hydroxychloroquine (HCQ) as a positive control; however, it has been revoked to treat COVID-19 by the U.S. FDA. Why the authors did not consider different candidates such as chloroquine, which also has endowed with S1R affinity.

Answer: The reviewer is right, HCQ was proposed for COVID-19, although it was later revoked, as we stated in the manuscript. Since it is known to bind S1R and it was proposed for COVID-19infectious treatment, we considered HCQ as an intriguing candidate to be further investigated, to better define its mode(s) of action.

The structurally related chloroquine is more toxic and less safe respect to HCQ and accordingly HCQ holds greater promise for repurposing/repositioning programs.

  1. 2. In Fig 2, the result displayed that ticlopidine obviously induced ATF4 and CHOP in both U87 and LNCaP cancer cells. However, the survival rate of U87 cells was the highest among the four tested cancer cells after the treatment of ticlopidine. Both data seems to contradict each other.

Answer:  We took into consideration the reviewer’s comment and added a brief discussion on this point in the text, as follows:

Ticlopidine exerts greater cytotoxicity in LNCaP cell line respect to U87 cell line, representative of glioblastoma multiforme, a type of aggressive brain tumor largely resistant to current treatments. Ticlopidine appears to circumvent inherent apoptosis resistance of U87 cells by inducing terminal UPR, even if with a lesser extent respect to LNCaP cells. This result is in line with the high sensitivity of LNCaP cells to endoplasmic reticulum (ER) stressors, which is reported in literature [1].

  1. Lindner P, r Christensen SB, Nissen P, Møller JV, Engedal N.Cell death induced by the ER stressor thapsigargin involves death receptor 5, a non-autophagic function of MAP1LC3B, and distinct contributions from unfolded protein response components. Cell Communication and Signaling; 18(1):12 2020

Reviewer 2 Report

Interesting paper looking at the role of ticlopidine for targeting ER stress after viral infection. 

Please provide update regarding potential utility for neurologic injury PMID: 28077004.

Also, please specify what part of the ER stress pathway is being targeted Bip is upstream and ATF4 and CHOP can be two different arms. Need clarification on exactly what part of the pathway is being targeted and potential implications for apoptosis PMID: 25540611

If points adequately addressed and above references included, could be of interest to the readership. 

Author Response

  1. Please provide update regarding potential utility for neurologic injury PMID: 28077004.

Answer: We thank the reviewer and we will take into account these observation for the future work. The therapeutic potential of ER stress modulation in neurological injury is a very interesting topic, but it is beyond the scope of this brief communication, as the therapeutic application we are focusing on are viral infections and cancer. Nevertheless, we included a short comment on the topic with a couple of references:

ER stress and sustained UPR signaling are significant contributors to the pathogenesis of several diseases, including inflammatory disorders, viral infections, neuronal degeneration and brain injury and can increase these events' severity [1,2].

[1] Lucke-Wold BP, Logsdon AF, Turner RC, Huber JD, Rosen CL. Endoplasmic Reticulum Stress Modulation as a Target for Ameliorating Effects of Blast Induced Traumatic Brain Injury. J Neurotrauma. 2017 Sep;34(S1):S62-S70. doi: 10.1089/neu.2016.4680.

[2] Yin Y, Sun G, Li E, Kiselyov K, Sun D. ER stress and impaired autophagy flux in neuronal degeneration and brain injury. Ageing Res Rev. 2017 Mar;34:3-14. doi: 10.1016/j.arr.2016.08.008. 

  1. Also, please specify what part of the ER stress pathway is being targeted Bip is upstream and ATF4 and CHOP can be two different arms. Need clarification on exactly what part of the pathway is being targeted and potential implications for apoptosis PMID: 25540611.

Answer: We took into consideration the reviewer’s comment and added the following part in the text:

Based on the data reported in the present work, we hypothesized that S1R inactivation by ticlopidine induces a very fast accumulation of unfolded proteins in ER lumen triggering the terminal UPR response. Actually, the protein chaperone BiP, and the transcription factors ATF4 and CHOP belong to the same pathway. Under normal physiological conditions, BiP is associated with ER sensor PERK present in ER membrane in an inactive state. As a response to the accumulation of unfolded or misfolded proteins within ER lumen, BiP dissociates from PERK, which becomes an active kinase with the ability to phosphorylate α subunits of the eIF2. The result is suppression of global protein translation, the cell cycle arrest in the G1 phase as well as the induction of preferential translation of ATF4, which upregulates expression of genes responsible for restoring cell homeostasis. Under prolonged ER stress, ATF4 trigger pro-apoptotic signals through induction of CHOP, which is responsible for initiation of the apoptotic cascade [2]. These evidence highlight once again the crucial role of the sigma 1receptors in controlling ER stress [3] and the activity of ticlopidine as ER stressor.

  1. RozpÄ™dek W, Pytel D, Mucha B, LeszczyÅ„ska H,  Diehl JA, Majsterek I. The Role of the PERK/eIF2α/ATF4/CHOP Signaling pathway in tumor progression during endoplasmic reticulum stress. Curr Mol Med. 2016; 16(6): 533–544.
  2. Tesei A, Cortesi M, Zamagni A, Arienti C, Pignatta S, Zanoni M, Paolillo M, Curti M; Rui M, Rossi D, Collina S. Sigma receptors as edoplasmic reticulum stress "gatekeepers" and their modulators as emerging new weapons in the fight against cancer. Frontiers in Pharmacology 2018. Front Pharmacol. 2018 Jul 10;9:711

Round 2

Reviewer 2 Report

Addressed